# Bacterial Infections among Patients with Chronic Diseases at a Tertiary Care Hospital in Saudi Arabia

**DOI:** 10.3390/microorganisms10101907

**Published:** 2022-09-26

**Authors:** Abdulrahman S. Bazaid, Ahmed A. Punjabi, Abdu Aldarhami, Husam Qanash, Ghaida Alsaif, Hattan Gattan, Heba Barnawi, Bandar Alharbi, Abdulaziz Alrashidi, Abdulaziz Alqadi

**Affiliations:** 1Department of Medical Laboratory Science, College of Applied Medical Sciences, University of Ha’il, Hail 55476, Saudi Arabia; 2Microbiology Unit, Department of Laboratory Medicine and Pathology BB, International Medical Center, Jeddah 21451, Saudi Arabia; 3Department of Medical Microbiology, Qunfudah Faculty of Medicine, Umm Al-Qura University, Al-Qunfudah 21961, Saudi Arabia; 4Molecular Diagnostics and Personalized Therapeutics Unit, University of Ha’il, Hail 55476, Saudi Arabia; 5Department of Medical Laboratory Technology, Faculty of Applied Medical Sciences, King Abdulaziz University, Jeddah 21589, Saudi Arabia; 6Special Infectious Agents Unit, King Fahad Medical Research Center, Jeddah 22252, Saudi Arabia; 7Security Forces Hospital Program, Riyadh 11481, Saudi Arabia

**Keywords:** bacterial infections, chronic diseases, antibiotic-resistant, diabetes mellitus and skin burns

## Abstract

Infections caused by multi-drug-resistant bacteria in patients with chronic diseases have been associated with high mortality and morbidity. While few reports have evaluated bacterial infections in multiple chronic disease patients, the focus of the current study was to investigate the prevalence of bacterial infections and the susceptibility profiles of causative strains among various groups of patients suffering from chronic diseases. Microbiological reports of patients suffering from cancer, diabetes mellitus, cardiovascular diseases, kidney diseases, and skin burns were retrospectively collected from a tertiary hospital in Saudi Arabia. Approximately 54.2% of recruited patients were males, and positive urine was the most prevalent specimen associated with kidney disease patients (25%)*. Escherichia coli* isolates were predominant among cardiovascular, kidney, and cancer patients. *Staphylococcus aureus* was commonly detected in diabetics and those with burns. Although resistance patterns varied based on the type of specimens and underlying diseases, *Escherichia coli* showed limited resistance to colistin, carbapenems, and tigecycline, while *S. aureus* demonstrated susceptibility to ciprofloxacin, gentamicin, and rifampin. These observations are crucial for clinicians and policymakers to ensure effective treatment plans and improve outcomes in these patients with comorbidity.

## 1. Introduction

Chronic disease can be defined as an illness that lasts for at least one year with continuous need for medical care and/or limited access to everyday activities [1]. Chronic diseases can be infectious (e.g., Hepatitis C) [2] and non-infectious, such as cancer, diabetes mellitus (DM), cardiovascular diseases (CVDs), and kidney disorders, although multiple non-infectious chronic conditions are initiated by infections, such as cervical cancer by human papillomavirus (HPV) [1]. In addition, recent investigations have suggested that burns should be considered a chronic disease since they can lead to a permanent reduction in the host’s immunity [3]. These chronic diseases are prevalent globally, although higher morbidity rates are linked to low-income and/or developing countries, including Saudi Arabia [4].

Cancer is one of the most prevalent chronic diseases worldwide, including in Saudi Arabia, where the annual incidence rate between 2010 and 2019 reached 17,522 (8296 men and 9226 women) per year [5]. Cancer patients generally are immune suppressed; therefore, they are more prone to various bacterial infections [6]. This, in turn, affects body parts with tumors or malignancies and may lead to death [6]. A previous study has claimed that patients with solid organ tumors died mainly because of complications resulting from infections [7]. Cancers tend to cause organ dysfunction, which increases the probability of contracting infections [6]. For example, patients whose central nervous systems are affected by tumors are frequently predisposed to various infections, such as peptic ulcers, pneumonia, osteomyelitis, and urinary tract infections (UTIs) [8]. This might be due to the use of immunosuppressant therapies (e.g., chemotherapy and biological response modifiers) that can impair the immune system of cancer patients leading to a higher rate of bacterial infections [9]. A study conducted in Saudi Arabia has reported that the majority of cancer patients with bloodstream infections (BSIs) were caused by Gram-negative bacteria, including *Klebsiella pneumonia* and *Escherichia coli* [10]. Thus, individuals with cancer require immediate and effective treatment for infections caused by drug-resistant bacteria.

The World Health Organization (WHO) has ranked Saudi Arabia as the second and the seventh highest country in the Middle East and the world for incidence of diabetes, respectively [11,12]. It was estimated that around 7 million of the Saudi population are diabetic, and almost 3 million showed signs of prediabetes [11]. A cross-sectional study conducted in Saudi Arabia revealed a higher rate of diabetes among males (34.1%) compared to females (27.6%) [13]. Another similar investigation was carried out in Jeddah, Saudi Arabia, in which about 1350 adult individuals with an average age of 45 years old were recruited, which showed that 9.0% of male and 8.6% of female participants are suffering from diabetes [14]. However, the incidence of DM and prediabetes increases with age since half of the participants over 50 years are diabetic [13]. In addition, patients with DM commonly complain of respiratory tract infections (RTIs), UTIs, gastrointestinal (GI) infections, and skin and soft tissue infections (SSTIs) [15]. This might be due to the high levels of sugar in the blood, tissues, and urine of patients with DM, allowing bacteria to grow and propagate rapidly, leading to infection [16]. In addition, high blood sugar indirectly affects the immune system of DM patients by impairing phagocytosis or/and reducing appropriate blood supply to blood vessels, which enhances the establishment of bacterial infections [16].

Patients with kidney diseases account for almost 6% of Saudi Arabia’s population and are at higher risk of various infections, including UTIs, pyelonephritis, and cystitis [17]. *Escherichia coli* was the predominant biological cause of UTIs [18,19]. *Staphylococcus* species can be translocated from the bloodstream and access the kidneys and urinary tract, causing severe complications [20]. Moreover, chronic diseases can overlap as diabetic patients are at high risk of having renal dysfunction and require hemodialysis; almost 40% of dialysis individuals are diabetic [21]. A recent study on tertiary care hospitals in Saudi Arabia observed that infections associated with bloodstream catheters occurred at a rate of 2.9 per 1000 days, and *Staphylococcus* species were the most common causative organisms [22]. Cardiovascular patients are also at high risk of infections because of invasive procedures used, including transplantation, congenital heart disease, and coronary artery bypass grafts. *Staphylococcus aureus*, particularly MRSA and *Acinetobacter baumannii,* are among the common bacteria associated with cardiovascular patients [23].

It has recently been reported that burns might be a major cause of skin infections [24]. *Staphylococcus aureus*, *Beta-hemolytic Streptococcus* group A, and *Enterococcus* species are the most common Gram-positive bacteria that infect burn wounds [24]. Damage to the skin caused by burns enhances bacteria’s ability to colonize wounds and establish local wound infection [24]. This local infection can then be disseminated to other organs or systems of the body, resulting in BSIs and complications [25]. A previous study on this matter in Saudi Arabia revealed that skin burns were the most prevalent illness among recruited patients with chronic diseases [26].

All the above-mentioned facts about patients with chronic diseases and their low immunity and high susceptibility to bacterial infections highlight the importance of investigating the prevalence and susceptibility profiles of bacteria that are associated with these patients, especially MDR bacteria, to limit their infections and associated complications. This can be achieved by providing effective and immediate treatment for infections caused by MDR bacteria targeting patients with chronic diseases. Thus, the aim of the current study is to determine and compare the prevalence of bacterial infections and antibiotic susceptibility profiles of causative strains among patients with chronic diseases.

## 2. Materials and Methods

### 2.1. Study Designs and Data Collection

This is a retrospective cross-sectional study conducted at King Salman Specialist Hospital (500 beds), Hail, Saudi Arabia. This hospital is a tertiary care hospital designated for patients with chronic diseases, including cancer, heart, and kidney illnesses, which usually require long-term care. Only data belonging to patients with chronic diseases (e.g., cancer, DM, CVDs, kidney diseases, and burns) who showed positive bacterial culture were included in this investigation. All relevant data, including diagnosis, age, gender, weight, height, and isolated bacteria and their resistance profiles, were collected anonymously over a period of 8 months (August 2020–April 2021). Patients with duplicate isolates or identical patterns were excluded from the study.

### 2.2. Bacterial Infection and Antimicrobial Susceptibility

Microbiological investigations for causative agents of potential infection in various samples requested from patients, including sputum, urine, blood, and wound swabs, were conducted, as explained previously [27]. Blood samples were collected in designated blood culture bottles (Becton, Dickinson and Franklin Lakes, NJ, USA) and incubated at 37 °C for 5 days using the BD BACTEC™ FX blood culture system (Becton, Dickinson and Franklin Lakes, NJ, USA). Sputum and wound samples were plated in blood and MacConkey agar plates. Urine samples were cultured on blood and cystine lactose electrolyte deficient (CLED) agar plates (Oxoid, Basingstoke, UK). All plates were incubated at 37 °C for 24 h. Bacterial identification was performed using multiple automated systems, including the BD Phoenix system (BD Biosciences, Franklin Lakes, NJ, USA) and MicroScan plus (Beckman Coulter, Brea, CA, USA). Susceptibility testing was evaluated using the BD BACTEC system. Gram-negative bacterial isolates were tested for various antibiotics, including amikacin, amoxicillin/clavulanic acid, ampicillin, aztreonam, cefepime, ceftazidime, cefoxitin, ceftriaxone, cefuroxime, cephalothin, ciprofloxacin, colistin, ertapenem, gentamicin, imipenem, meropenem, nitrofurantoin, Levofloxacin, tigecycline, trimethoprim/sulfamethoxazole, and piperacillin-tazobactam. Sensitivity profiles of Gram-positive isolates were determined against amoxicillin/clavulanic acid, ampicillin, cefoxitin, ciprofloxacin, clindamycin, gentamicin, erythromycin, oxacillin, penicillin, tetracycline, trimethoprim/sulfamethoxazole, and rifampin. However, to ensure accuracy and reproducibility, antibiotic susceptibility was confirmed by the disk diffusion method according to the Clinical and Laboratory Standard Institute [28].

### 2.3. Statistical Analysis

Obtained data, including patient demographics, bacterial identities, and antibiotic resistance profiles, were analyzed using GraphPad Prism version 9.3.0. Analyzed data are presented as number of cases and percentages to provide an accurate and reproducible representation of tested populations. In addition, the inferential statistical test (*t*-test) was conducted to assess differences between variables in which a *p*-value < 0.05 was considered statistically significant and vice versa.

### 2.4. Ethical Approval

The study design was reviewed and approved by the Ethics Committee at Hail Affairs (reference: H-08-L-074). A consent form was not required because this was a retrospective study with no interaction with patients. Patient privacy and confidentiality of data were maintained anonymously in accordance with The Declaration of Helsinki.

## 3. Results

The current study highlighted the prevalence and antibiotic susceptibility among patients with chronic disease. A total of 96 patients were included in the study; 55 (54.2%) were male and 44 (45.8%) were female (Table 1). When compared to other underlying diseases, kidney disease was more reported in female (40.9%) than male (15.4%) patients (Table 1). It was also found that patients with DM represented 32.3% of the participants, while patients with kidney diseases, burns, CVDs, and cancer were 27.1%, 18.8%, 14.6%, and 7.3%, respectively (Table 1). Interestingly, 63 (65.6%) participants were older than 50 years, and 24 (38.1%) of these subjects were primarily diagnosed with DM, followed by 17 (27.0%) with kidney diseases, 13 (20.6%) with CVDs (20.0%), 6 (9.5%) with cancer, and 3 (4.8%) subjects with burns (Table 1). Based on body mass index (BMI) calculation, 53 (55.2%) participants were normal, and only 43 (44.8%) were abnormal, including 2 (2.1%) and 41 (42.7%) patients who were underweight and obese, respectively (Table 1).

In the current study, a total of 96 Gram-positive and Gram-negative bacterial isolates were detected in all recruited patients with various underlying diseases (Table 2). Generally, a notable increase in the number of cases caused by Gram-negative bacteria (75.0%) was observed compared to incidents belonging to Gram-positive bacteria (25.0%) among all patients and their various investigated clinical specimens (Table 2). Notably, *K. pneumoniae* (25%) and *E. coli* (25%) were the most dominant bacterial isolates among Gram-negative and/or all isolated bacteria from patients with chronic disease, followed by *A. baumannii* (9.4%), while *S. aureus* (15.6%) was substantially the most prevalent Gram-positive pathogen among all detected Gram-positive bacteria from patients with chronic diseases (Table 2). Irrespective of tested clinical specimens from recruited patients, the highest portion of isolates (32.3%) was detected in patients suffering from DM, followed by kidney (27.1%) and burn patients (18.8%), while the least percentage of isolates was observed in CVD (14.6%) and cancer (7.3%) patients (Table 2). Nevertheless, the majority of positive bacterial cultures, regardless of the bacterial identity, were revealed to be for urine samples among kidney patients (25%), followed by wound samples from either burns (18.8%) or DM (17.7%) patients (Table 2). In contrast, only one bacterial isolate was identified in the sputum sample of DM patients and in wound and blood samples of patients with kidney disease (Table 2). In addition, bacteremia due to either Gram-positive or Gram-negative bacterial isolates was mostly observed in patients with DM (3.1%), followed by CVD (2.1%) and kidney (1%) patients. However, sputum infections were only reported in CVD (4.2%) and DM (1%) patients (Table 2).

## 4. Discussion

A microbial infection is a life-threatening event that can lead to serious complications, such as sepsis and/or multiple organ dysfunction syndrome (MODS), and death, especially in patients with chronic diseases, including cancer, DM, CVDs, kidney diseases, and burns [29]. This may be due to multiple reasons, mainly the weak immunity of individuals with underlying diseases and the growing tragedy of antimicrobial resistance (AMR) [8]. Studies have reported bacterial infections among specific populations, such as diabetic, cancer, or cardiovascular patients; however, there are limited/absent reports that have evaluated bacterial infections among multiple chronic disease patients simultaneously to compare their overall prevalence and antibiotic resistance profiles. Thus, the focus of the current study was to investigate the prevalence of bacterial infections and the susceptibility profiles of causative strains among patients suffering from chronic diseases, including cancer, DM, CVDs, kidney diseases, and burns to allow quantitative comparisons and assessment between individuals with these illnesses. In addition, the susceptibility profiles of bacteria associated with patients with chronic diseases will be essential to enhance the therapeutic choice and effectiveness of prescribed antibiotics, especially empirical drugs [18,27].

Overall, data showed that the highest positive bacterial cultures of wound specimens were observed for patients with burns (45%), while kidney patients were revealed to be associated with the highest positive bacterial cultures of urine samples (53%) compared to all other clinical samples. This finding confirms that individuals with dysfunctional skin and kidneys are more prone to wound and urinary tract infections, respectively [30]. Nevertheless, DM patients showed the highest positive bacterial cultures of both wound and urine samples (combined) (32%) compared with other patients. Diabetic patients tend to have dry skin, frequent urination, high levels of blood sugar (hyperglycemia), and poor immunity and blood circulation, which explains their higher susceptibility to contracting skin and urinary tract infections in comparison with other patients [16]. Akash et al. argue that the immune-compromised state of DM patients caused by hyperglycemia is the main factor leading to increased bacterial infections in DM patients [31]. This is because hyperglycemia caused by DM has been linked to decreased humoral immunity and delayed tissue recuperation, making individuals more prone to secondary bacterial infections, especially wounds [31]. In addition, DM is a more predominant illness compared with other chronic diseases in the kingdom of Saudi Arabia [13]. However, the complexity of bacterial infections in DM patients depends on the severity, stages of diabetes, and individual immune response [32]. In this context, the progression of diabetes is well-linked with the risk of bacterial infection and its complications [32].

Positive sputum culture is well-linked with patients with CVDs followed by diabetic individuals, and a similar trend was observed for positive blood cultures, although no positive culture from other patients was documented, except a positive blood sample from a kidney patient. The vulnerability of patients with DM, CVDs, and kidney diseases to respiratory infections and bacteremia might be due to various factors that may be overlapped somehow, including poor immunity, use of injections (e.g., insulin and anticoagulant injections), and other medical devices (e.g., dialysis machine) [32]. Additionally, the higher rate of bacteremia or/and septicemia among DM patients was attributable to glycemic control measures and medications [31]. Thus, all the above-mentioned reasons should explain why the current data has revealed DM patients as being more susceptible to bacterial infections than patients with other chronic diseases (cancer, CVDs, kidney disease, and burns). This finding concurs with former studies addressing the prevalence of bacterial infections in chronically ill patients [24,31].

Obtained data showed that Gram-negative bacteria were more prominent than Gram-positive bacteria among each group of patients as well as the total tested population. This could be due to the higher level of pathogenicity for most Gram-negative bacteria resulting from its sophisticated structure, production of various invasive toxins, and resistance to immune responses (e.g., phagocytosis) and antibiotics [33]. However, this observation can be prone to bias as the type of bacteria (Gram-positive or negative) can be influenced by the type of tested specimens (urine, wound, sputum, and blood) and underlying disease (cancer, DM, CVDs, kidney disease, and burns) of diagnosed patients. This is because urine samples in the current study are found to be more associated with Gram-negative bacteria (*Escherichia coli* and *Klebsiella pneumoniae*), whereas Gram-positive bacteria (*Staphylococcus aureus*) are mostly identified from wound samples. This variation of bacterial ability to establish infection targeting different human systems (e.g., urinary tract and skin) may be mainly due to different virulence factors and natural habitats of causative bacteria [34]. Nevertheless, the general level of host immunity that is related to each human system can be a real contributing factor to the difference between Gram-positive and negative bacteria to cause infection in multiple human systems. Thus, patients suffering from kidney illness are more prone to urinary tract infections (UTIs), while wound infections are more prevalent among patients with burns.

Similar resistance profiles of *E. coli* strains isolated from all patients included in this investigation were observed, although strains of this bacterium associated with cancer patients showed the highest level of resistance compared to strains of other patients. Thus, bacterial infections among cancer patients (solid and/or hematological malignancies), especially caused by MDR bacteria, are serious complications leading to a high mortality rate [35,36,37]. This can be mainly attributed to their impaired immunity. Although *E. coli* strains isolated from urine specimens of cancer patients possess higher levels of resistance compared with other detected *E. coli* strains, a previous investigation revealed that *K. pneumoniae* is the most predominant bacterium, highly resistant, and associated with mortality among patients with tumors [38]. In addition, MDR *A. baumannii* was reported as a prevalent bacterium among cancer patients [39], but this is not the case in the current data as there was not even a single isolate of *A*. *baumannii* detected from all specimens of cancer patients. This variation might be due to the limited number of recruited patients with cancer compared with other studies. Thus, recruiting a larger population of cancer patients would be a rational task to validate this observation.

Strains of *K. pneumoniae* followed by *E. coli* and then *Staphylococcus aureus* are more prevalent in urine and wound (combined) samples of DM patients. Strains of *K. pneumoniae* isolated from wound specimens of DM patients showed a wider residence profile toward tested antibiotics, including colistin and tetracycline drugs, while other strains of this bacterium from urine specimens of diabetic and kidney patients showed lower resistance patterns and full sensitivity towards colistin and tetracycline antibiotics. This high pattern of resistance by strains of *K. pneumoniae* could suggest a strong correlation between MDR bacteria and DM patients in comparison with other obtained resistance profiles and patients of chronic diseases. This observation is consistent with a previous report which claimed the relationship between DM patients and MDR bacteria, such as MDR bacteria, causes UTIs [40].

Cardiovascular diseases (CVD) are the cause of morbidity for 8.9 million people every year worldwide [41]. These rates are reported to be more than deaths caused by cancer, respiratory diseases, and HIV combined [41]. There are cases of CVD reported as a complication of bacterial infections, such as diphtheria, that contribute to the global burden of myocarditis [42]. On the other hand, few data reported bacterial infection as a complication in patients with CVD except when associated with DM [43]. However, hospitalization of CVD patients, such as patients with heart failure, is associated with higher bacterial infection rates. Bacteria such as *Clostridium difficile* were found higher in hospitalized patients diagnosed with heart failure compared with those without this illness [44]. In this study, only 14% of the positive bacterial culture were isolated from CVD patients. These data indicate that CVD patients are less susceptible to infection compared to patients with DM, kidney disease, and burns. *A. baumannii K. pneumoniae, E. coli, S. aureus, and E. faecium* are all isolated from CVD patients, among which *E. coli* was the most dominant bacteria isolated from urine samples. Isolated *E. coli* strains from patients with CVD showed variable levels of resistance to tested antibiotics, with the highest level of resistance to ampicillin. Bacterial infection in CVD patients could have resulted from nosocomial infections after prolonged hospitalization or/and admission into the intensive care unit (ICU) [45]. The current study also revealed that CVD and DM patients had more bacteremia than other recruited patients. Studies have shown that bacteremia can lead to sepsis, which carries an increased risk of cardiovascular complications, such as myocardial infarction, acute heart failure, or stroke [46].

The majority of bacterial infections among kidney patients were caused by *K. pneumoniae* and *E. coli,* followed by *Acinetobacter baumannii, Pseudomonas aeruginosa,* and *Serratia marcescens* which were less common. This finding is in parallel with other studies, which showed that patients with chronic kidney disease (KD) are at higher risk of infection with different bacteria, especially bloodstream infections [47]. The most common bacterial infections associated with KD were *E. coli* and *S. aureus* [48]. Other possible infections are caused by Gram-negative and -positive bacteria, including *Streptococcus pneumoniae*, *Enterococcus species*, and *K. pneumoniae* [47]. Another study revealed that different bacteria might be isolated from renal disease patients, such as *S. aureus, E. coli, Pseudomonas aeruginosa, and Klebsiella* species [49]. In addition, antibiotic-resistant isolates, including methicillin-resistant *Staphylococcus aureus* (MRSA) and vancomycin-resistant enterococci (VRE), can increase the rate of morbidity and mortality in KD patients [49].

Burn injuries are considered a chronic disease since it leads to an increased risk of secondary pathologies, such as cancer, CVDs, and microbial infections [3]. *S. aureus* (33%) was the most common bacterium among patients with burns. *S. aureus* is well-known as the main cause of wound infections, including MRSA, which is the predominant wound bacterium in the ICU [50]. *Staphylococcus* species harbor several virulence factors, including proteinases, collagenases, and hyaluronidase, that enhance bacterial invasion and dissemination, leading to systemic infection and septic shock [51]. Although over half of patients with burns have been reported to be colonized with MRSA, only a single MRSA strain was detected in the current investigation. A possible explanation is that most recruited patients in this study were admitted or/and treated in the emergency department, not in the ICU. Although *S. aureus* in patients with burns showed a higher resistance rate to certain antibiotics, such as ampicillin and erythromycin, it was fully susceptible to clindamycin and tetracycline. Clindamycin can be used as an empirical antibiotic in treating hospitalized patients with SSTIs, including burns [52]. Nevertheless, *P. aeruginosa* (22%) and *A. baumannii* (22%) were also commonly observed among patients with burns. *P. aeruginosa* is not only limited to a local infection in patients with burns (wound infection), but it can disseminate via the bloodstream leading to bacteremia, sepsis, and death [24]. Wounds and burns are moist environments that are suitable for the establishment and propagation of *P. aeruginosa* [53]. Infections caused by *P. aeruginosa* are usually treated by piperacillin-tazobactam due to the resistance patterns of this bacterium [54]. In the current study, strains of *P. aeruginosa* showed intermediate resistance patterns to piperacillin-tazobactam and meropenem and were sensitive to colistin. However, MDR *P. aeruginosa* has been reported; thus, regular monitoring of the resistance pattern of this bacterium is crucial to adjust the drug of choice [55]. Although colistin is active against *P. aeruginosa* and most other Gram-negative bacteria, it has been associated with toxicity, especially in the kidney and lung [56]. Thus, prolonged use of this antibiotic should be avoided, particularly for individuals with kidney diseases, and close monitoring is highly recommended when colistin is prescribed. In addition, the second most prevalent isolated Gram-negative bacteria among patients with burns were *A. baumannii* (22%). This bacterium has the ability to live in harsh environments (dry) and has been strongly associated with immunocompromised patients [57]. Traditionally *A. baumannii* is susceptible to carbapenems (e.g., imipenem and meropenem) [57], but in the current study, the strains of this bacterium showed full resistance to all tested antibiotics except colistin, which is the only active drug to eradicate infections caused by these strains of *A. baumannii*. In the context of bacterial infections and antibiotic resistance among patients with chronic diseases, it is worth mentioning the high possibility of mixed bacterial cultures or bacterial co-infection in these patients leading to serious complications resulting mainly from treatment failure. The availability of at least two different bacterial species simultaneously could interact with each other to enhance their invasion of the host (pathogenicity) by multiple means, including biofilm formation that is highly resistant to antibiotic drugs. Thus, co-infections and biofilm formation are two linked processes that can be more prevalent in patients with incompetent immunity, which requires an effective combination of antibiotics [58].

## 5. Conclusions

Bacterial infections, especially those caused by MDR bacteria in patients with chronic diseases, could be a life-threatening event since they carry a high rate of resistance that can lead to treatment failure, serious complications, and death. *E. coli* isolates exhibited a low resistance profile, being fully susceptible to colistin, imperium, and meropenem, whereas strains of *A. baumannii* isolated from wound samples belonging to patients with burns showed the highest level of resistance to all tested antibiotics except colistin. *S. aureus* exhibited good susceptibility to ciprofloxacin, gentamicin, and rifampin. This data will help local doctors to prescribe effective drugs for treating infections in patients with chronic diseases to limit/control any possible serious complications and reduce the mortality rate. The current study reported bacterial prevalence and antibiotic susceptibility profiles among chronically ill patients. Frequent monitoring and surveillance studies for the resistance pattern of bacteria that are commonly associated with patients with chronic diseases are recommended.

## Figures and Tables

**Figure 1 microorganisms-10-01907-f001:**
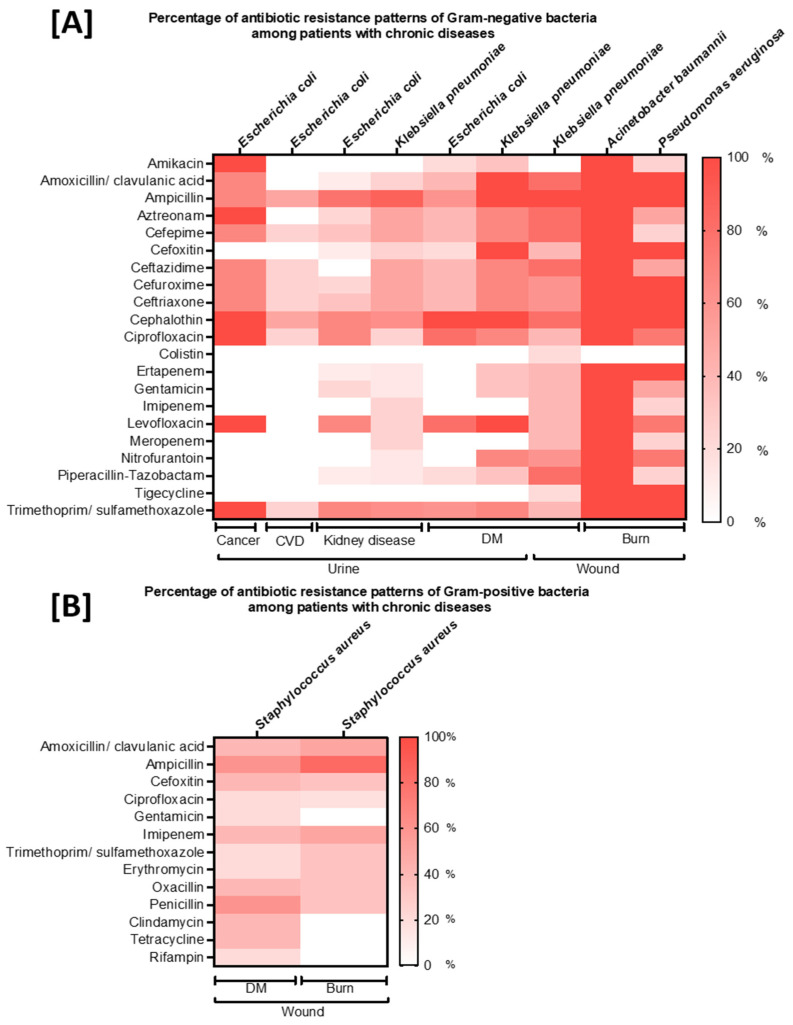
Heat maps of antibiotic resistance profiles. (**A**) Resistance profile of Gram-negative bacteria (*Escherichia coli, Klebsiella pneumoniae, Acinetobacter baumannii*, and *Pseudomonas aeruginosa*) associated with patients suffering from cancer, cardiovascular diseases, kidney disease, diabetes mellitus, and burn) that were detected from urine and wound samples. (**B**) Resistance profiles of Gram-positive bacteria (*Staphylococcus aureus)* associated with patients suffering from diabetes mellitus (DM) and burns that were detected from wound specimens. Results are presented based on the red color gradient reflecting the antibiotic resistance rate. The very intense/strong red color indicates the highest (100%) rate of resistance, whereas the absence of the red color was used for the lowest (0%) rate of resistance.

**Table 1 microorganisms-10-01907-t001:** Demographic characteristics of the study cohort, a total of 96 patients divided based on their underlying diseases and compared by their gender, age, and body mass index (BMI); results presented in number and (percentage).

	Cancer	DM	CVDs	Kidney Diseases	Burns	Total
**Gender**						
** Male**	3 (6)	18 (35)	8 (15)	8 (15)	15 (29)	52
** Female**	4 (9)	13 (29)	6 (14)	18 (41)	3 (7)	44
**Age**						
** 1–10**	-	-	-	2 (67)	1 (33)	3
** 11–20**	-	1 (25)	-	2 (50)	1 (25)	4
** 21–30**	-	1 (33)	-	1 (33)	1 (33)	3
** 31–40**	-	4 (27)	-	1 (7)	10 (67)	15
** 41–50**	1 (12)	1 (12)	1 (12)	3 (38)	2 (25)	8
** >50**	6 (9)	24 (38)	13 (21)	17 (27)	3 (5)	63
**BMI**						
** Underweight (>18.5)**	-	1 (50)	-	1 (50)	-	2
** Normal (18.8–29.5)**	5 (9)	19 (36)	6 (11)	10 (19)	13 (25)	53
** Obese (<30)**	2 (5)	11 (27)	8 (19)	15 (37)	5 (12)	41
**Total**	**7**	**31**	**14**	**26**	**18**	**96**

DM: diabetes mellitus; CVDs: cardiovascular diseases.

**Table 2 microorganisms-10-01907-t002:** Identified bacteria of positive bacterial culture tested specimens. Bacterial isolates (*n* = 96) were divided into Gram-negative and Gram-positive bacteria, collected from different sites, including urine, wound, sputum, and blood, according to each represented disease (cancer, DM, CVD, kidney disease, and burn).

Microbial Isolates	Cancer		DM				CVDs				Kidney Diseases			Burns	Total	*p*-Value
	Urine	Wound	Urine	Wound	Sputum	Blood	Urine	Wound	Sputum	Blood	Urine	Wound	Blood	Wound		
**Gram (-) bacteria**																
** *Acinetobacter baumannii* **	-	-	-	-	-	1	-	-	1	-	2	-	1	4	9	*
** *Escherichia coli* **	3	-	5	2	-	-	4	-	-	1	9	-	-	-	24	****
** *Klebsiella pneumoniae* **	2	-	3	5	-	-	2	1	2	-	8	-	-	1	24	****
** *Proteus mirabilis* **	-	-	-	1	-	-	-	-	-	-	-	-	-	2	3	ns
** *Pseudomonas aeruginosa* **	-	-	-	-	-	-	-	-	-	-	2	1	-	4	7	****
**Others**	-	1	-	2	-	-	-	-	-	-	2	-	-	-	5	ns
**Gram (+) bacteria**																
** *Enterobacter faecalis* **	-	-	2	1	-	-	-	-	-	-	-	-	-	1	4	ns
** *Staphylococcus aureus* **	-	1	-	5	-	1	-	1	-	1	-	-	-	6	15	****
**Others**	-	-	-	1	1	1	-	-	1	-	1	-	-	-	5	ns
**Total**	**5**	**2**	**10**	**17**	**1**	**3**	**6**	**2**	**4**	**2**	**24**	**1**	**1**	**18**	**96**	

DM: diabetes mellitus; CVDs: cardiovascular diseases; statistical significance: *p*-value of <0.05 (*), <0.0001 (****). Others in Gram-negative include *Serratia marcescens, Citrobacter koseri, Proteus vulgaris,* and *Klebsiella oxytoca.* Others in Gram-positive include *Enterococcus faecium, Enterobacter cloacae,* and *Enterobacter aerogenes*. A total of 21 antibiotics were used to measure the resistance of bacteria that were isolated from patients with chronic diseases (Figure 1). In comparison to Gram-positive bacterial isolates, a higher percentage of antibiotic resistance was reported in Gram-negative bacteria isolated from DM and burn patients. Remarkably, all bacterial isolates detected from urine and wound specimens of patients suffering from cancer, CVDs, kidney diseases, DM, and/or burns showed reduced resistance to colistin (Figure 1A). Notably, strains of *Acinetobacter baumannii* isolated from patients with burns were monitored with the highest percentage of resistance as it has presented with complete resistance to all tested antibiotics except colistin (Figure 1A). A variable level of resistance was observed for Gram-positive bacteria isolated from patients with DM, kidney diseases, cancer, or CVDs (Figure 1B). Interestingly, strains of *S. aureus* identified in patients with DM and burns showed a high level of sensitivity to almost all tested antibiotics, but an intermediate level of resistance to some antibiotics was observed, such as amoxicillin, penicillin, and ampicillin (Figure 1B).

## Data Availability

All data are available within the manuscript.

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
