# Peer review of "Bacterial Infections among Patients with Chronic Diseases at a Tertiary Care Hospital in Saudi Arabia"

_microorganisms, 2022, doi:10.3390/microorganisms10101907_

Round 1

Reviewer 1 Report

Please add at the end of the introduction a clear aim of the work

It’s better to describe in the method section some details about the process of antimicrobial sensitivity  

it will be added value for the manuscript if you include a section about the about presence of MDR gene in the detected bacteria 

it will be added value if added a section describing the examination the virulence gene presence in the detected bacteria 

Author Response

We would like to thank you for reviewing our manuscript. Comments are always helpful, and we appreciate the efforts taken to improve our manuscript. The response to each point raised has been detailed below.

  1. Please add at the end of the introduction a clear aim of the work

  Response: This was addressed.

  1. It’s better to describe in the method section some details about the process of antimicrobial sensitivity  

Response: This was addressed and more information were added.

  1. it will be added value for the manuscript if you include a section about the about presence of MDR gene in the detected bacteria 

Response: We totally agree with this comment as such information will definitely increase the value of the manuscript. Nevertheless, hospital/s where the data collected from are not performing genetic screening for resistant genes, so we would be unable to provide this information. 

  1. it will be added value if added a section describing the examination the virulence gene presence in the detected bacteria 

Response: We appreciate and agree with this comment since discussing bacterial virulence genes would increase the value of this manuscript. However, because hospital/s where the data collected from are not performing genetic screening for resistant/virulence genes, we are unable to provide this information. 

Reviewer 2 Report

The article has a scientific and practical interest. The study was conducted on a large sample of patients with chronic diseases for  analyses concomitant bacterial infections. The resistance of the identified bacteria to a set of antibiotics was experimentally studied. Increased resistance to antibiotics in bacteria causing nosocomial infections has been shown. The authors collected and analyzed quite a lot of material, which indicates on the reliability of the conclusion. The results are presented quite concisely. The discussion gives clear comments and basic possibilities to compare obtained results with the data of other researchers. The results may be useful when considering the spread of nosocomial bacterial infections in chronic patients, as well as for assessing the risk of associated infections. In addition, information on the resistance of the most common nosocomial bacteria is useful in developing a strategy to combat bacterial infections in chronic patients.

Some notes.

1) Keywords do not adequately reflect the content of the article. Thus, they do not mention bacterial infections, although their study was one of the objectives of this work.

2) The description of the methods of bacteriological research is very brief. This concerns the identification of the obtained isolates, the conditions of their cultivation.

3) Figure 1 shows two heat maps, although there is only one caption for the figure. Figures should be marked with letters a and b and reflect this in the caption to the figure.

4) In the discussion, it would be interesting to raise the issue of the possibility of developing mixed bacterial infections in chronic patients and related processes of biofilm formation, which may show increased resistance to antibiotics.

5) Since the bacterial infections considered in the article have been studied quite well, it is desirable to more clearly indicate the novelty of the results obtained in the discussion and conclusion.

Author Response

We would like to thank you for reviewing our manuscript. Comments are always helpful, and we appreciate the efforts taken to improve our manuscript. The response to each point raised has been detailed below.

  1. Keywords do not adequately reflect the content of the article. Thus, they do not mention bacterial infections, although their study was one of the objectives of this work.

Response: This was addressed.

  1. The description of the methods of bacteriological research is very brief. This concerns the identification of the obtained isolates, the conditions of their cultivation.

Response: This was addressed as more details regarding the identification of the obtained isolates and their conditions of cultivation were added to the Method section.  

  1. Figure 1 shows two heat maps, although there is only one caption for the figure. Figures should be marked with letters a and b and reflect this in the caption to the figure.

Response: Figures was revised. font size was corrected, titles and letters (A and B) to each figure were added as requested.

  1. In the discussion, it would be interesting to raise the issue of the possibility of developing mixed bacterial infections in chronic patients and related processes of biofilm formation, which may show increased resistance to antibiotics.

Response: A section about mixed bacteria and biofilm formation and how this can be serious event in patients with chronic dieses due to the high rate of resistance was added to the discussion section.

  1. Since the bacterial infections considered in the article have been studied quite well, it is desirable to more clearly indicate the novelty of the results obtained in the discussion and conclusion.

Response: A clear novelty statement was added to the abstract, discussion and conclusion of the manuscript. Although studies have reported bacterial infections among a specific population such as diabetic, cancer or cardiovascular patients, limited/absent reports that have evaluated bacterial infections among multiple chronic disease patients simultaneously to compare their overall infection rate (prevalence) and antibiotic resistance profiles.

Reviewer 3 Report

The present manuscript focuses on the Bacterial Infections among Patients with Chronic Diseases at a 2

Tertiary Care Hospital in Saudi Arabia.  The subject frame of the work is well constructed. So, in this respect and this article should be contributed to present research. I recommended this work for publication after the following minor revisions.

1.      There are several typographical mistakes as well in whole manuscript. Therefore, the author’s thoroughly careful check the language and typo mistake to minimize the error.

2.      The abstract should be beginning with a sentence about the background of concept and the aims as well as novelty of study should be mentions. What exactly is the novelty of this study? The abstract is poorly written and should be improved. Abbreviations must be avoided in abstract. Parenthesis should be avoided in abstract - this is poor writing. Please improve.

3.      The introduction and discussion section about infection and bacteria in hospital need extensive revision and improved. Be specific and adhere to importance of topic.

4.      All figures are of poor technical quality and not suitable for publication, especially in a high reputed journal. Font size and kind is too small and must be unified in all figures. Small writings are unreadable. All figures must be self-explanatory. Axis titles are poorly presented or absent. Units are missing. Are the data presented in figures significantly different? At least error bars should be shown.

5.      What is exactly the novelty of this article, as so many articles were already out. Author needs to revised it carefully and should provide novelty statement.

6.      Bacteria species/genus name should be italic throughout the manuscript.

7.      I suggest to provide a pictorial mechanism of whole study at the end of discussion. 

Author Response

We would like to thank you for reviewing our manuscript. Comments are always helpful, and we appreciate the efforts taken to improve our manuscript. The response to each point raised has been detailed below.

  1. There are several typographical mistakes as well in whole manuscript. Therefore, the author’s thoroughly careful check the language and typo mistake to minimize the error.

Response: The entire manuscript was revised as requested.

  1. The abstract should be beginning with a sentence about the background of concept and the aims as well as novelty of study should be mentions. What exactly is the novelty of this study? The abstract is poorly written and should be improved. Abbreviations must be avoided in abstract. Parenthesis should be avoided in abstract - this is poor writing. Please improve.

Response: The abstract was revised and improved as requested.

  1. The introduction and discussion section about infection and bacteria in hospital need extensive revision and improved. Be specific and adhere to importance of topic.

Response: The section about bacteria in hospital was removed as the main focus of the study was bacterial infection in patients with chronic diseases rather than hospitalized patients.

  1. All figures are of poor technical quality and not suitable for publication, especially in a high reputed journal. Font size and kind is too small and must be unified in all figures. Small writings are unreadable. All figures must be self-explanatory. Axis titles are poorly presented or absent. Units are missing. Are the data presented in figures significantly different? At least error bars should be shown.

Response: Figures were revised. font size was corrected and title to each figure and unit were added. Regarding the absence of significant difference (statistical analysis), the t-test is performed on absolutes frequencies only, while our results are presented in overall percentages for both heat maps. As this analysis would not be possible conduct upon our data, we have followed the color gradient method to present the highest and/or lowest rate of resistance among tested bacteria to highlight the difference between overall resistance profiles. 

  1. What is exactly the novelty of this article, as so many articles were already out. Author needs to revised it carefully and should provide novelty statement.

Response: The novelty statement was added to abstract, discussion and conclusion. Although studies have reported bacterial infection among specific population such as diabetic, cancer or cardiovascular patients, limited reports evaluated bacterial infection in different chronic disease patients and correlate their antibiotic resistant profiles.

  1. Bacteria species/genus name should be italic throughout the manuscript.

Response: This was corrected throughout the manuscript. 

  1. I suggest to provide a pictorial mechanism of whole study at the end of discussion. 

Response: We have added a pictorial mechanism summarizing the main findings of the study as requested. We would appreciate that this figure considers as graphical abstract.